# PREDICTIVE UNCERTAINTY THROUGH QUANTIZATION

## ABSTRACT

High-risk domains require reliable confidence estimates from predictive models. Deep latent variable models provide these, but suffer from the rigid variational distributions used for tractable inference, which err on the side of overconfidence. We propose Stochastic Quantized Activation Distributions (SQUAD), which imposes a flexible yet tractable distribution over discretized latent variables. The proposed method is scalable, self-normalizing and sample efficient. We demonstrate that the model fully utilizes the flexible distribution, learns interesting non-linearities, and provides predictive uncertainty of competitive quality.

## 1 INTRODUCTION

In high-risk domains, prediction errors come at high costs. Luckily such domains often provide a fail-safe: self-driving cars perform an emergency stop, doctors run another diagnostic test, and industrial processes are temporarily halted. For deep learning models, this can be achieved by rejecting datapoints with a confidence score below a predetermined threshold. This way, a low error rate can be guaranteed at the cost of rejecting some predictions. However, estimating high quality confidence scores from neural networks, which create well-ordered rankings of correct and incorrect predictions, remains an active area of research.

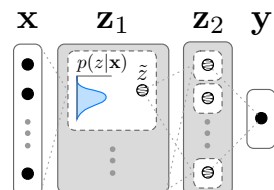

Figure 1: DLVMs have layers of stochastic latent variables.

Deep Latent Variable Models (DLVMs, fig. 1) approach this by postulating latent variables $\mathbf{z}$ for which the uncertainty in $p(\mathbf{z}|\mathbf{x})$ influences the confidence in the target prediction. Recently, efficient inference algorithms have been proposed in the form of variational inference, where an inference neural network is optimized to predict parameters of a variational distribution that approximates an otherwise intractable distribution (Kingma & Welling (2013); Rezende et al. (2014); Alemi et al. (2016); Achille & Soatto (2016)).

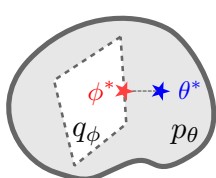

Figure 2: A distribution $q_\phi$ is optimized to approximate $p_{\theta^*}$.

Variational inference relies on a tractable class of distributions that can be optimized to closely resemble the true distribution (fig. 2), and it's hypothesized that more flexible classes lead to more faithful approximations and thus better performance (Jordan et al. (1999)). To explore this hypothesis, we propose a novel tractable class of highly flexible variational distributions. Considering that neural networks with low-precision activations exhibit good performance (Holi & Hwang (1993); Hubara et al. (2016)), we make the modeling assumption that latent variables can be expressed under a strong quantization scheme, without loss of predictive fidelity. If this assumption holds, it becomes tractable to model a scalar latent variable with a flexible multinomial distribution over the quantization bins (fig. 3).

By re-positioning the variational distribution from a potentially limited description of moments, as found in commonly applied conjugate distributions, to a direct expression of probabilities per value, a variety of benefits arise. As the output domain is constrained, the method becomes self-normalizing, relieving the model from hard-to-parallelize batch normalization techniques (Ioffe & Szegedy (2015)). More interesting priors can be explored and the model is able to learn unique activation functions per neuron.

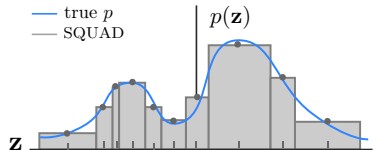

Figure 3: SQUAD quantizes the domain of $z$ to model a flexible and tractable variational distribution.

More concretely, the contributions of this work are as follows:

- We propose a novel variational inference method by leveraging multinomial distributions on quantized latent variables.
- We show that the emerging predicted distributions are multi-modal, motivating the need for flexible distributions in variational inference.
- We demonstrate that the proposed method applied to the information bottleneck objective computes competitive uncertainty over the predictions and that this manifests in better performance under strong risk guarantees.

## 2 BACKGROUND

In this work, we explore deep neural networks for regression and classification. We have data-points consisting of inputs $\mathbf{x}$ and targets $\mathbf{y}$ in a dataset $\mathcal{D} = \{(\mathbf{x}^i, \mathbf{y}^i) \mid i \in [1, ..., N]\}$ and postulate latent variables $\mathbf{z}$ that represent the data. We focus on the Information Bottleneck (IB) perspective: first proposed by Tishby et al. (2000), the information bottleneck objective $I(\mathbf{y}, \mathbf{z}; \theta) - \beta I(\mathbf{x}, \mathbf{z}; \theta)$ is optimized to maximize the mutual information between $\mathbf{z}$ and $\mathbf{y}$, whilst minimizing the mutual information between $\mathbf{z}$ and $\mathbf{x}$. The objective can be efficiently optimized using a variational inference scheme as shown concurrently by both Alemi et al. (2016) and Achille & Soatto (2016). Under the Markov assumption $P(\mathbf{z}, \mathbf{x}, \mathbf{y}) = p(\mathbf{z}|\mathbf{x})p(\mathbf{y}|\mathbf{x})p(\mathbf{x})$, they derive the following lower bound:

$$I(\mathbf{y}, \mathbf{z}; \theta) - \beta I(\mathbf{x}, \mathbf{z}; \theta) \geq \mathcal{L} = \frac{1}{N} \sum_{n=1}^{N} \mathbb{E}_{p_\theta(\mathbf{z}|\mathbf{x}_n)}[\log q_\theta(\mathbf{y}_n|\mathbf{z})] - \beta D_{\mathrm{KL}}[p_\theta(\mathbf{z}|\mathbf{x}_n)\|r(\mathbf{z})], \quad (1)$$

where $\mathbb{E}_{p(\mathbf{z}|\mathbf{x}_n)}$ is commonly estimated using a single Monte Carlo sample and $r(\mathbf{z})$ is a variational approximation to the marginal distribution of $\mathbf{z}$. In practice $r(\mathbf{z})$ is fixed to a simple distribution such as a spherical Gaussian. Alemi et al. (2016) and Achille & Soatto (2016) continue to show that the Variational Auto Encoder (VAE) Evidence Lower Bound (ELBO) proposed in Kingma & Ba (2014); Rezende et al. (2014) is a special case of the IB bound when $\mathbf{y} = \mathbf{x}$ and $\beta = 1$:

$$I(z, x) - \beta I(z, i) \geq \mathbb{E}_{p_\theta(\mathbf{z}|\mathbf{x}_n)}[\log q_\theta(\mathbf{x}_n|\mathbf{z})] - D_{\mathrm{KL}}[p_\theta(\mathbf{z}|\mathbf{x}_n)\|r(\mathbf{z})], \quad (2)$$

where $i$ represents the identity of data-point $\mathbf{x}_i$. Interestingly, the VAE perspective considers the bound to optimize a variational distribution $q(\mathbf{z}|\mathbf{x})$, whilst the IB perspective prescribes that $q(\mathbf{z}|\mathbf{x})$ in the ELBO is not a variational posterior but the true encoder $p(\mathbf{z}|\mathbf{x})$, and instead $p(\mathbf{y}|\mathbf{z})$ and $p(\mathbf{z})$ are the distributions approximated with variational counterparts. However, deriving the ELBO on a conditional distribution $p(\mathbf{y}|\mathbf{x})$ leads to a KL divergence between the approximate posterior $q(\mathbf{z}|\mathbf{x})$ and the real posterior $p(\mathbf{z}|\mathbf{x})$, which is not available at test time, rather than the prior $p(\mathbf{z})$. Hence, the information bottleneck perspective provides a more fundamental motivation for the bound in equation 1.

Alternatively, equation 1 can be interpreted as a domain-translating beta-VAE (Higgins et al. (2016)), where an input image is encoded into a latent space and decoded into the target domain. The Lagrange multiplier $\beta$ then controls the trade-off between rate and distortion, as argued by Alemi et al. (2017). However, this does not naturally arise from deriving the ELBO using Jensen's inequality on the conditional likelihood $p(\mathbf{y}|\mathbf{x})$.

## 3 METHOD

In this work, we follow the IB interpretation of the bound in equation 1 and leave the evaluation of our proposed variational inference scheme in other models such as the VAE for further work. At the heart of our proposal lies the assumption that neuron networks can be effective under strong activation quantization schemes. We start with presenting the derivation if the model in the context of a single latent-layer information bottleneck, following the single data-point loss in equation 1 and dropping the subscript $_n$ for clarity, with figure 4 for visual reference:

$$\mathcal{L} = \mathcal{L}_{\mathrm{error}} - \mathcal{L}_{\mathrm{KL}} = \mathbb{E}_{p_\theta(\mathbf{z}|\mathbf{x})}[\log q_\theta(\mathbf{y}|\mathbf{z})] - \beta D_{\mathrm{KL}}[p_\theta(\mathbf{z}|\mathbf{x})\|r(\mathbf{z})]. \quad (3)$$

To impose a flexible, multi-modal distribution over $\mathbf{z}$, we first make a mean-field assumption $p(\mathbf{z}|\mathbf{x}) = \prod_k p(z_k|\mathbf{x})$. We then quantize the *domain* of each of the $K$ scalar latent variables $z_k$

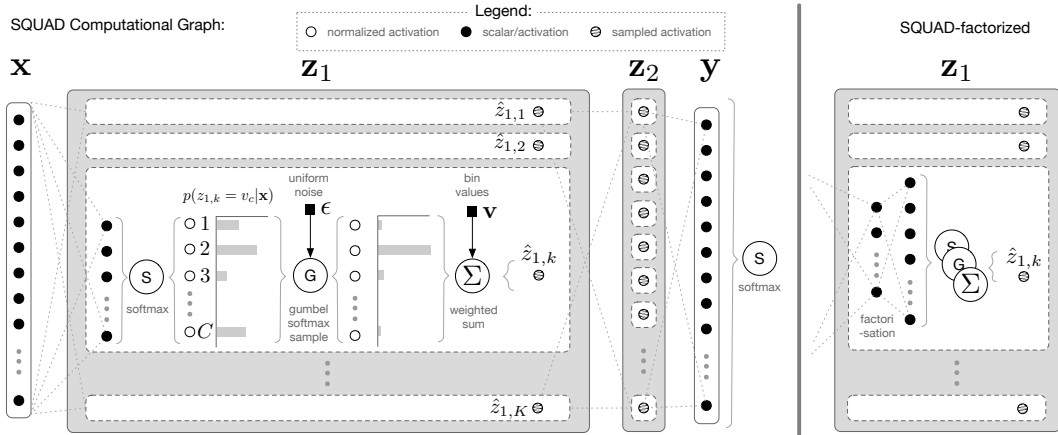

Figure 4: The left diagram visualizes the computational graph of SQUAD at training time, providing a detailed view on how an individual latent variable is sampled. The right diagram visualizes how the proposed matrix-factorization variant improves the parameter efficiency of the model.

such that only a small set of potential values remain: $z_k \in \mathbf{v} = \{v_1, \ldots, v_C\}$ $\forall k$ with e.g. $C = 11$, see fig. 3. We use the same quantization scheme, defined by $\mathbf{v}$, for all the latents. The specific scheme is considered part of the prior, and is discussed in figure 5.

Although the number of values of $\mathbf{z}$ is relatively small, the space of values over all the latent variables grows exponentially. We thus take to a monte-carlo estimation scheme. To optimize the parameters $\theta$ with Stochastic Gradient Descent (SGD), we need to derive a fully differentiable sampling scheme that allows us to sample values of $\mathbf{z}$. . To formulate this, we re-parametrize the expectation over $\mathbf{z}$ in equation 3 using a set of variables $s_k \in \{1, \ldots, C\}$ which index the value vector $\mathbf{v}$, allowing us to use a softmax function to represent the distribution over $\mathbf{z}$:

$$p_\theta(z_k = v_c|\mathbf{x}) \triangleq p_\theta(s_k = c|\mathbf{x}) = \text{softmax}(\mathbf{W}\mathbf{x} + \mathbf{b})_c. \tag{4}$$

These indexing values $\mathbf{s}$ are then used in conjunction with values $\mathbf{v}$ as in input for $q_\theta(\mathbf{y}|\mathbf{z})$, which is modelled with a small network $f_\theta(\cdot)$: (*abusing notation to indicate element-wise indexing with* $\mathbf{v}[\mathbf{s}]$):

$$\mathcal{L}_{\text{error}} = \mathbb{E}_{\mathbf{s} \sim p_\theta}[\log f_\theta(\mathbf{v}[\mathbf{s}])].$$

To enable sampling from the discrete variables $\mathbf{s}$, we use the Gumbel-Max trick (Gumbel (1954)), denoted $\text{gumb}()$, re-parameterizing the expectation $\mathbb{E}_{\mathbf{s} \sim p}$ with uniform noise $\epsilon \sim U(0, 1)$:

$$\mathcal{L}_{\text{error}} = \mathbb{E}_\epsilon \left[ \log f_\theta \left( \mathbf{v} \left[ \arg\max_c \text{gumb}(p_\theta(\mathbf{s}|\mathbf{x}), \epsilon) \right] \right) \right]. \tag{5}$$

As the argmax is not differentiable, we approximate this expectation using the Gumbel-Softmax trick (Maddison et al. (2016); Jang et al. (2016)), which generates samples that smoothly deform into one-hot samples as the softmax temperature $\tau$ approaches 0. Using the inner product (denoted ·) of the approximate one-hot samples and $\mathbf{v}$, we create samples from $\mathbf{z}$:

$$\mathcal{L}_{\text{error}} \approx \mathbb{E}_\epsilon \left[ \log f_\theta \left( \mathbf{v} \cdot \text{softmax}_c \text{gumb}(p_\theta(\mathbf{s}|\mathbf{x}), \epsilon) \right) \right]. \tag{6}$$

In practice, we anneal $\tau$ from $1.0$ to $0.5$ during the training process, as proposed by Yang et al. (2017) to reduce gradient variance initially, at the risk of introducing bias.

To conclude our derivation, we use a fixed SQUAD distribution to model the variational marginal $r(\mathbf{z})$ as shown in figure 5. We can then derive the KL term analytically following the definition for discrete distributions. Using the fact that the KL divergence is additive for independent variables,

we get our final loss:

$$\mathcal{L} = \mathbb{E}_\epsilon \left[ \log f_\theta \left( \mathbf{v} \cdot \operatorname*{softmax}_c \operatorname{gumb}(p_\theta(\mathbf{s}|\mathbf{x}), \epsilon) \right) \right] - \beta \sum_{k=1}^{K} \sum_{c=1}^{C} p_\theta(s_k = c|\mathbf{x}) \log \frac{p_\theta(s_k = c|\mathbf{x})}{r(z_k = v_c)}. \tag{7}$$

For the remainder of this work, we will refer to the latent variables as $\mathbf{z}$ in lieu of $\mathbf{s}$, for clarity.

At test time, we can approximate the predictive function $p(y^*|\mathbf{x}^*)$ for a new data-point $\mathbf{x}^*$ by taking $T$ samples from the latent variables $\mathbf{z}$ i.e. $\hat{\mathbf{z}}_t \sim p(\mathbf{z}|\mathbf{x}^*)$, and averaging the predictions for $y^*$:

$$p_\theta(y^*|\mathbf{x}^*) \approx \int q_\theta(y^*|\mathbf{z}) p_\theta(\mathbf{z}|\mathbf{x}^*) d\mathbf{z} \approx \frac{1}{T} \sum_{t=1}^{T} q_\theta(y^*|\hat{\mathbf{z}}_t). \tag{8}$$

A downside of mean-field variational approximations is that latents are assumed to be uncorrelated. We can bring some correlation to the latents, and extend the flexibility of the proposed model, by creating a hierarchical set of latent variables. We maintain a mean-field assumption for the prior, and the joint posterior distribution of L layers of latents factorizes as follows:

$$p_\theta(\mathbf{z}_1, \ldots, \mathbf{z}_L|\mathbf{x}) = p_\theta(\mathbf{z}_L|\mathbf{z}_{L-1}) \cdots p_\theta(\mathbf{z}_1|\mathbf{x}), \tag{9}$$

With $q_\theta(\mathbf{y}|\mathbf{z}_1, \ldots, \mathbf{z}_L) = q_\theta(\mathbf{y}|\mathbf{z}_L)$. This is straightforwardly implemented with a simple ancestral sampling scheme.

Interestingly, the strong quantization proposed in our method can itself be considered an additional information bottleneck, as it exactly upper-bounds the number of bits per latent variable. Such bottlenecks are theorized to have a beneficial effect on generalization (Tishby et al. (2000); Achille & Soatto (2016); Alemi et al. (2017; 2016)), and we can directly control this bottleneck by varying the number of quantization bins.

The computational complexity of the method, as well as the number of model parameters $\theta$, scale linearly in $C$, i.e. $O(C)$ (with $C$ the number of quantization bins). It is thus suitable for large-scale inference. We would like to stress that the proposed method differs from work that leverages the Gumbel-Softmax trick to model categorical latent variables: our proposal models continuous scalar latent variables by quantizing their domain and modeling belief with a multinomial distribution. Categorical latent variable models would incur a much larger polynomial complexity penalty of $O(C^2)$.

The method is easily integrated into existing deep neural network architectures, and SQUAD layer implementations are provided at `github.com/anonymized`.

**Matrix-factorization variant**   In the natural image domain, we anticipate a need for a high amount of information per variable, for which a small number of bins does not suffice. To improving the tractability of using a large number of quantization bins, we propose a variant of SQUAD that uses a matrix factorization scheme to improve the parameter efficiency. Formally, equation 4 becomes:

$$p(s_k = c|\mathbf{x}) = \operatorname{softmax}(\mathbf{w}''_{k,c}(\mathbf{w}'_k \mathbf{x} + \mathbf{b}'_k) + \mathbf{b}''_{k,c}),$$

with full layer weights $\mathbf{W}''$ and $\mathbf{W}'$ respectively of shape $(K, B, C)$ and $(|X|, K, B)$, where $K$ denotes the number of neurons, $B$ the number of factorization inputs, $C$ number of quantization bins and $|X|$ the input dimensionality. To improve the parameter efficiency, we can learn $\mathbf{W}'$ per layer as well, resulting in shape $(|X|, 1, B)$, which is found to be beneficial for large C by the hyperparameter search presented in section 5. We depict this alternative model on the right side of figure 4 and will refer to it as SQUAD-factorized. We leave further extensions such as Network-in-Network (Lin et al. (2013)) for future work.

## 4   RELATED WORK

Outside the realm of DLVMs, other methods have been explored for predictive uncertainty. Lakshminarayanan et al. (2017) propose deep ensembles: straightforward averaging of predictions from a

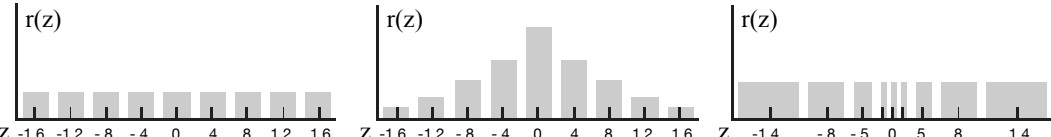

Figure 5: In the IB bound, the marginal $p(z)$ is approximated with a fixed distribution $r(z)$. Using our proposed SQUAD distribution we can impose a variety of interesting forms for $r(\mathbf{z})$ via the spacing $\mathbf{v}$ and weighting $r(z_k = v_c)$ of the quantization bins. For the values $\mathbf{v}$, we compare linearly spaced bins **(left)** versus bins with equal probability mass under a normal distribution **(right)**. Furthermore, we explore the effect of allowing the bin values to be optimized with SGD on a per-neuron or per-layer basis, to allow the model to optimize the quantization scheme with the highest fidelity. For the prior probabilities, we explore a uniform prior **(left)** and probability mass of the bins under a normal distribution **(middle)**.

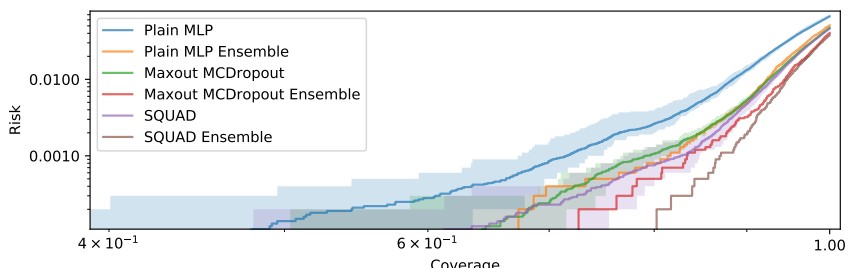

Figure 6: Risk/coverage curve (with log-axes for discernibility) of 2-layer models on notMNIST. Lines closer to the lower-right are better. As the selective classifier lowers the confidence threshold, coverage increase at the cost of greater classification risk. Area around curves represents 90% confidence bounds computed using 10 initializations/splits.

small set of separately adversarially trained DNNs. Although highly scalable, this method requires retraining a model up to 10 times, which can be inhibitively expensive for large datasets.

Gal & Ghahramani (2015b) propose the use of dropout (Srivastava et al. (2014)) at test time (Monte Carlo Dropout, or MC-Dropout) and present a Bayesian neural network interpretation of this method. A follow-up work by Gal et al. (2017) explores the use of Gumbel-Softmax to smoothly deform the dropout noise to allow optimization of the dropout rate during training. A downside of MC-Dropout (MCD) is the limited flexibility of the fixed bi-modal delta-peak distribution imposed on the weights, which requires a large number of samples for good estimates of uncertainty. van den Oord et al. (2017) propose the use of vector quantization in variational inference, quantizing a multi-dimensional embedding, rather than individual latent variables, and explore this in the context of auto-encoders.

In the space of learning non-linearity's, Su et al. (2017) explore a flexible non-linearity that can assume the form of most canonical activations. More flexible distributions have been explored for distributional reinforcement learning by Dabney et al. (2017) using quantile regression, of which can be seen as a special case of SQUAD where the bin values are learned but have fixed uniform probability. Categorical distributions on scalar variables have been used to model more flexible Bayesian neural network posteriors as by Shayer et al. (2017). The use of a mixture of diracs distribution to approximate a variety of distributions was proposed by Schrempf et al. (2006).

## 5 RESULTS

Quantifying the quality of uncertainty estimates of models remains an open problem. Various methods have been explored in previous works, such as relative entropy Louizos & Welling (2017); Gal & Ghahramani (2015a), probability calibration, and proper scoring rules Lakshminarayanan et al. (2017). Although interesting in their own right, these metrics do not directly measure a good ranking of predictions, nor indicate applicability in high-risk domains. Proper scoring rules are the exception, but a model with good ranking ability does not necessarily exhibit good performance on proper scoring rules: any score that provides relative ordering suffices and does not have to reflect true calibrated probabilities. In fact, well-ranked confidence scores can be re-calibrated (Niculescu-Mizil & Caruana (2005)) after training to improve performance on proper scoring rules and calibration metrics.

In order to evaluate the applicability of the model in high-risk fields such as medicine, we want to quantify how models perform under a desired risk guarantee. We propose to use the *Selection with Guaranteed Risk* (SGR) method[1] introduced by Geifman & El-Yaniv (2017) to measure this. In summary, the SGR method uses a confidence score, e.g. the estimated probability of the prediction, and rejects the prediction if this score is below a certain threshold. This threshold is optimized on a hold-out set to the tightest possible value that still guarantees a desired minimial error rate with high probability (e.g. 99%). SGR is formally proven, and empirically shown, to ensure this minimal error rate under i.i.d. assumptions. We propose to use SGR as evaluation metric for confidence estimation methods. Specifically, on a hold-out set we measure the percentage of predictions that are accepted by SGR under a specific risk guarantee, i.e. the **coverage**. To illustrate, one can imagine that a trivial solution to guarantee no errors is to reject all data-points, which has 0% coverage. By lowering the confidence threshold, coverage goes up, but the accuracy of the method goes down. SGR finds the optimal threshold that ensures the accuracy does not drop below a predetermined threshold, and by measuring the coverage under this threshold we can effectively compute the real world value of a confidence estimation methods.

To limit the influence of hyper-parameters on the comparison, we use a extensive hyper-parameter scheme on all variants and baselines. Specifically we use the automated optimization method TPE (Bergstra et al. (2011)) over a maximum of 1000 evaluations per model per experiment. The hyper-parameters are optimized for coverage at 2% risk ($\delta = 0.01$) on fashionMNIST (Xiao et al. (2017)), and we evaluate the optimal hyperparameters on another dataset, notMNIST, to measure how sensitive the hyperparameters are to a change in the data-distribution. Larger models are evaluated on Street View Housing Numbers (SVHN) (Netzer et al. (2011)) using the same optimization scheme. For the details on the evaluated search space and chosen hyper-parameters we refer the reader to the appendix.

We compare[2] our model against plain Multi-Layer Perceptrons (MLPs) with ReLU activations, MC-Dropout using Maxout activations[3] (Goodfellow et al. (2013); Chang & Chen (2015)) and an information bottleneck model using mean-field Gaussian distributions. We evaluate the complementary deep ensembles technique (Lakshminarayanan et al. (2017)) for all methods.

### 5.1 MAIN RESULTS

We start our analysis by comparing the predictive uncertainty of 2-layer models with 32 latent variables per layer. In figure 6 we visualize the risk/coverage trade-off achieved using the predicted uncertainty as a selective threshold, and present coverage results in table 1. Overall, we find that SQUAD performs significantly better than plain MLPs and deep Gaussian IB models, and we tentatively attribute this to the increased flexibility of the multinomial distribution. Compared to a Maxout

---

[1]We deviate slightly from Geifman & El-Yaniv (2017) in that we use Softmax Response (SR) – the probability taken from the softmax output for the most likely class – as the confidence score for all methods. Geifman & El-Yaniv (2017) instead proposed to use the variance of the probabilities for MCdropout, but our experiments showed that SR paints MCDropout in a more favorable light.

[2]All models are optimized with ADAM (Kingma & Ba (2014)), weight initialization as proposed by He et al. (2015), a weight decay of $10^{-5}$ and adaptive learning rate decay scheme — 10x reduction after 10 epochs of no validation accuracy improvement— and use early stopping after 20 epochs of no improvement.

[3]We found this baseline to perform stronger in comparison to conventional ReLU MCdropout models, under equal number of latent variables.

Table 1: SGR coverage results on Fashion MNIST. We present coverage percentage of the test dataset for three pre-determined risk-guarantees (one per column), where higher coverage is better, as well as negative log-likelihood and overall accuracy. The results indicate that SQUAD provides competitive uncertainty, especially at low-risk guarantees. Bayesian approximations via deep ensembles improve coverage all over the board, and for SQUAD in particular. When SGR can not guarantee the required risk level at high probability, 0% coverage is reported. (*2 std. deviations shown in parentheses, optimal results in **bold**.*)

| **Fashion MNIST** | cov@risk .5% | cov@risk 1% | cov@risk 2% | NLL | Acc. |
|---|---|---|---|---|---|
| Plain MLP | 29.1 (±20.71) | 45.9 (±4.04) | 60.4 (±3.17) | 0.408 (±.036) | 87.7 (±.42) |
| Maxout MCDropout | 41.9 (±9.86) | 56.5 (±2.30) | **69.9** (±1.48) | 0.299 (±.008) | **89.5** (±.28) |
| DLGM | 0.0 (±.00) | 33.5 (±2.42) | 47.0 (±1.47) | 0.446 (±.007) | 84.3 (±.15) |
| SQUAD | **42.9** (±7.19) | **58.3** (±3.06) | 69.5 (±1.55) | **0.293** (±.008) | **89.5** (±.35) |
| **Deep Ensemble** | cov@risk .5% | cov@risk 1% | cov@risk 2% | NLL | Acc. |
| Plain MLP Ensemble | 40.6 | 58.3 | 70.2 | 0.296 | 89.3 |
| Max. MCD. Ensemble | **48.2** | 59.1 | 72.2 | **0.271** | **90.2** |
| DLGM Ensemble | 0.0 | 34.3 | 47.8 | 0.435 | 84.7 |
| SQUAD Ensemble | 47.5 | **61.6** | **73.1** | 0.273 | 90.1 |

Table 2: SQUAD exhibits strong performance on notMNIST using the optimal hyperparameters found for fashionMNIST.

| **notMNIST** | cov@risk .5% | cov@risk 1% | cov@risk 2% | NLL | Acc. |
|---|---|---|---|---|---|
| Plain MLP | 77.4 (±5.20) | 85.5 (±1.91) | 90.3 (±.87) | 0.228 (±.009) | 93.3 (±.29) |
| Maxout MCDropout | 85.7 (±1.14) | 90.6 (±.62) | 94.2 (±.36) | 0.165 (±.003) | 95.3 (±.21) |
| SQUAD | **87.1** (±1.60) | **91.1** (±.86) | **94.5** (±.50) | **0.161** (±.006) | **95.4** (±.22) |
| Plain MLP Ensemble | 85.9 | 90.6 | 93.5 | 0.175 | 94.9 |
| Max. MCD. Ensemble | 88.5 | 92.8 | 95.7 | 0.148 | 96.0 |
| SQUAD Ensemble | **90.7** | **93.5** | **96.1** | **0.137** | **96.2** |

Table 3: Results on SVHN indicate that the quantization scheme imposed by SQUAD models might hinder performance, but that this is effectively compensated by the SQUAD-factorized variant using a larger amount of bins. Even with $T = 4$ MC samples at test time, SQUAD performs well.

| **MLP K=256** (SVHN) | cov@risk .5% | cov@risk 1% | cov@risk 2% | NLL | Acc. |
|---|---|---|---|---|---|
| Plain MLP | 0.0 (±.00) | 0.0 (±.00) | 36.3 (±2.84) | 0.758 (±.065) | 83.1 (±.42) |
| Maxout MCDropout | 0.0 (±.00) | 50.7 (±1.42) | 65.0 (±2.32) | 0.480 (±.020) | 86.4 (±.71) |
| SQUAD-factorized | **18.4** (±30.33) | **53.9** (±1.81) | **66.7** (±2.10) | **0.454** (±.021) | **86.7** (±.88) |
| SQUAD | 1.7 (±6.75) | 42.8 (±2.25) | 59.3 (±1.38) | 0.534 (±.005) | 84.6 (±.13) |
| Max. MCDropout T=4 | 0.0 (±.00) | 38.5 (±3.85) | 57.6 (±2.66) | 0.562 (±.016) | 84.9 (±.73) |
| SQUAD-factorized T=4 | **10.7** (±26.25) | **49.9** (±3.51) | **64.5** (±2.45) | **0.480** (±.024) | **86.2** (±.67) |
| SQUAD T=4 | 0.0 (±.00) | 38.0 (±1.86) | 55.6 (±.84) | 0.569 (±.019) | 83.7 (±.52) |

MCdropout model with a similar number of weights, SQUAD appears to have a slight —though not significant— advantage, despite the strong quantization scheme, especially at low risk guarantees. Deep ensembles improve results for all methods, which fits the hypothesis that ensembles integrate over a form of weight uncertainty. When comparing the optimal hyperparameters found for fashionMNIST on the similar notMNIST dataset, we find that SQUAD shows strong performance, as shown in table 2. This provides some evidence that despite the increase in number hyperparameters in SQUAD, the optimal settings are more robust to a change in the data distribution.

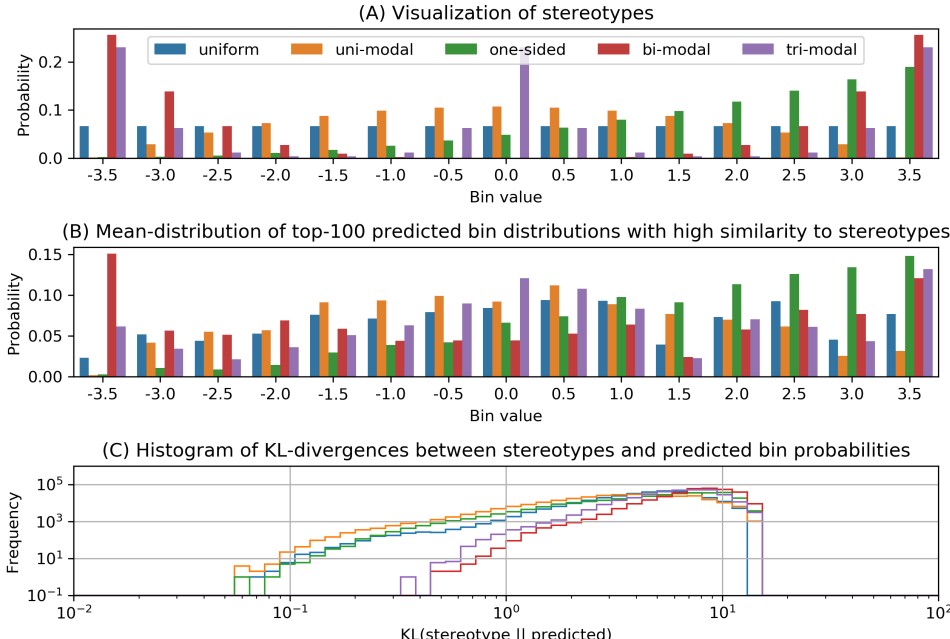

Figure 7: By analyzing the emerging predicted distributions of individual neurons in a converged SQUAD model, we find that the flexible variational distribution is used to its full advantage. Figure (A) visualizes a subset of interesting stereotypical distributions we hope to find in the model. Figure (B) summarizes distributions predicted by the model similar to stereotypes, discovered by looking at predicted distributions with low KL. Figure (C) shows how often distributions similar to stereotypes arise, as measured by the KL distance (lower KL is closer to stereotypes).

## 5.2 NATURAL IMAGES

To explore larger models trained on natural image datasets, we re-tune hyper-parameters on 256-latent 2-layer models over 100 TPE evaluations. As SVHN contains natural images in color, we anticipate a need for a higher amount of information per variable. We thus explore the effect of the matrix-factorized variant.

As shown in table 3, SQUAD-factorized outperforms the non-factorized variant. Considering the computational cost at the optimum of a 4-neuron factorization ($B = 4$) with $C$=37 quantization bins, the model clocks 3.4 million weights. In comparison, the optimum for the presented MCdropout results has $C$=11, using 9.0 million weights. On an NVIDIA Titan Xp, the dropout baseline takes 13s per epoch on average, while SQUAD-factorized spans just 9s.

To evaluate the sample efficiency of the methods, we compare results at $T = 4$ samples. We find that SQUAD's results suffer less from under-sampling the predictive distribution than MCdropout. We tentatively attribute the sample efficiency to the flexible approximating posterior on the activations, which is in stark contrast to the rigid approximating distribution that MCdropout imposes on the weights. In conclusion, SQUAD comes out favorably in a resource-constrained environment.

## 5.3 ANALYSIS OF LATENT VARIABLE DISTRIBUTIONS

In order to evaluate if the proposed variational distribution does not simply collapse into single mode predictions, we want to find out what type of distributions the model predicts over the latent variables. We visualize the forms of predicted distributions in figure 7a. Although this showcases only a small subset of potential multi-modal behavior that emerges, this demonstrates that the model indeed utilizes the distribution to its full potential. To provide an intuition on how these predicted distributions emerge, we present figure 9 in the appendix.

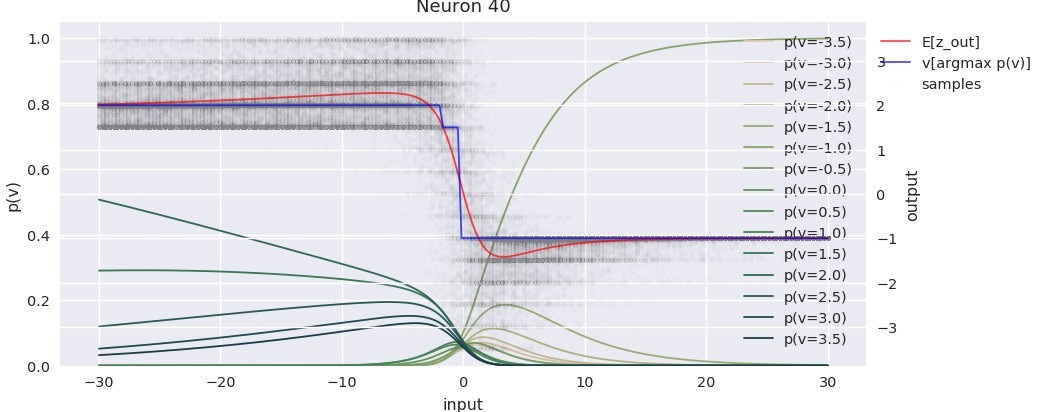

Figure 8: By using a 1-dimensional matrix factorization for a SQUAD-factorized distribution, we can visualize the type of (stochastic) activation functions learned by the method. After training the model as usual on fashion-MNIST, we take a random neuron from the first layer. We visualize how the predicted distribution of the output changes as a function of the 1-dimensional input. The left y-axis indicates the prob(ability per value as shown using the green line. The right y-axis indicates the value and in blue the most likely value is shown, and the gray dots represent samples from the neuron. The red line depicts the *expected output* of the neuron. The shape of the expected output is akin to a peaky sigmoid activation, and similar shapes are found in the other neurons of the network as well. This provides food for thought on the design of activation functions for conventional neural networks.

In figure 8 we visualize one of the activation functions that the method learns for a 1-dimensional input SQUAD-factorized model. The learned activation functions resemble "peaked" sigmoid activations, which can be interpreted as a combination of an RBF kernel and sigmoid. This provides food for thought on how non-linearity's for conventional neural networks can be designed, and the effect of using such a non-linearity can be studied in further work.

## 6 DISCUSSION

In this work, we have proposed a new flexible class of variational distributions. To measure the effectiveness for real world classification, we applied the class to a deep variational information bottleneck model. By placing a quantization-based distribution on the activations, we can compute uncertainty estimates over the outputs. We proposed an evaluation scheme motivated by the need in real-world domains to guarantee a minimal risk. The results presented indicate that SQUAD provides an improvement over plain neural networks and Gaussian information bottleneck models. In comparison to a MCDropout model, which approximates a Bayesian neural network, we get competitive performance. Moreover, qualitatively we find that the flexible distribution is used to its full advantage is sample efficient. The method learns interesting non-linearity's, is tractable and scale-able, and as the output domain is constrained, no batch normalization techniques are required.

Various directions for future work arise. The improvement of ensemble methods over individual models indicates that there remains room for improvement for capturing the full uncertainty of the output, and thus a fully Bayesian approach to SQUAD which would include weight uncertainty, shows promise. The flexible class allows us to define a wide variety of interesting priors, which provides opportunity to study interesting priors that are hard to define as a continuous density. Likewise, more effective initialization of parameters for the proposed method requires further attention. Orthogonally, the proposed class can be applied to other variational objectives as well, such as the variational auto-encoder. Finally, the discretized nature of the variables allows for the analytical computation of other divergences such as mutual information and the Jensen-Shannon divergence, the effectiveness of which remains to be studied.

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

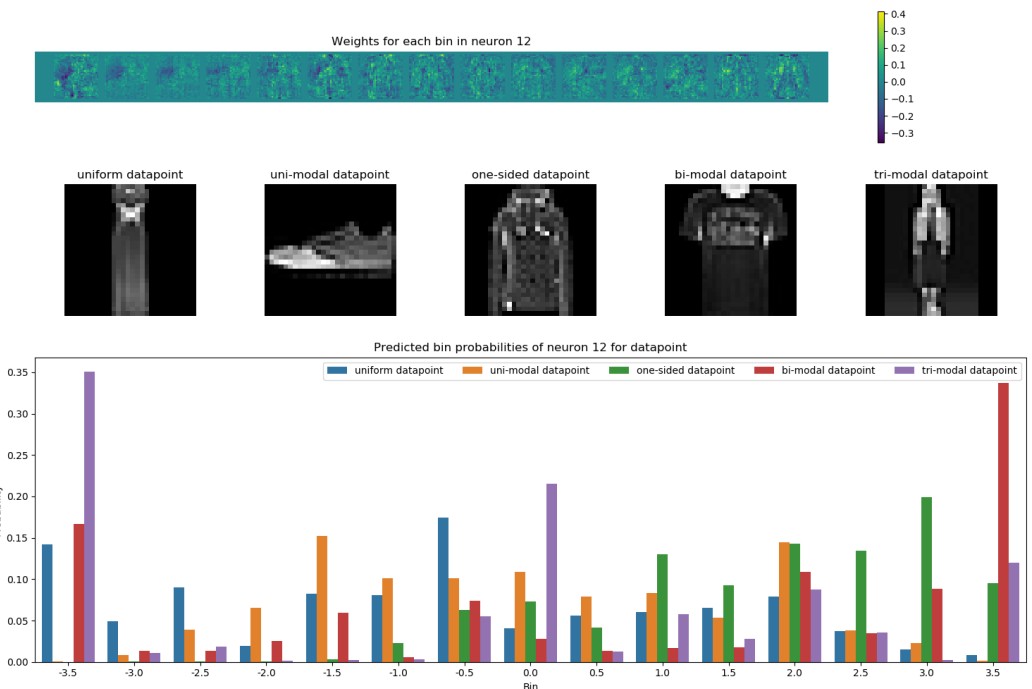

Figure 9: This figure serves to provide intuition on how a variety of distributions come about in our model. We show the set of weights used to predict the probability for the $C$ bins of a randomly selected latent variable $\mathbf{z}_{l=1,k=12}$ from the first layer in a converged 2-layer SQUAD model (reshaped to a 28x28 squares for comparison with the data). We then present 5 data-points for which the neuron predicts a stereotypical distribution, as visualized in the last bar-plot.

# 7 APPENDIX

## 7.1 EFFECT OF HYPER-PARAMETERS ON COVERAGE:

The optimal configuration of hyper-parameters and bin priors have been determined using 700 evaluations selected using TPE. The space of parameters explored is as follows, presented in the hyperopt API for transparency:

```
# Shared
C: quniform(2, 10, 1) * 2 + 1,
dropout rate: uniform(0.01, .95),
lr: loguniform(log(0.0001), log(0.01)),
batch_size: qloguniform(log(32), log(512), 1)
# SQUAD & Gaussian
kl_multiplier: loguniform(log(1e-6), log(0.01)),
init_scale: loguniform(log(1e-3), log(20)),
# SQUAD
use_bin_probs: choice(['uni', 'gaus']),
use_bins: choice(['equal_prob_gaus',
                   'linearly_spaced']),
learn_bin_values: choice([
   'per_neuron', 'per_layer', 'fixed']),
```

In figure 10 we visualize the pairwise effect of these hyper-parameters on the coverage. The optimal configuration found in for the main SQUAD model are: batch size: 244, KL multiplier: 0.0027, learn bin values: per layer, $p(z)$: uniform, **v**: linearly spread over (-3.5,3.5), lr: 0.0008, $C$: 15, initialization scale: 3.214.

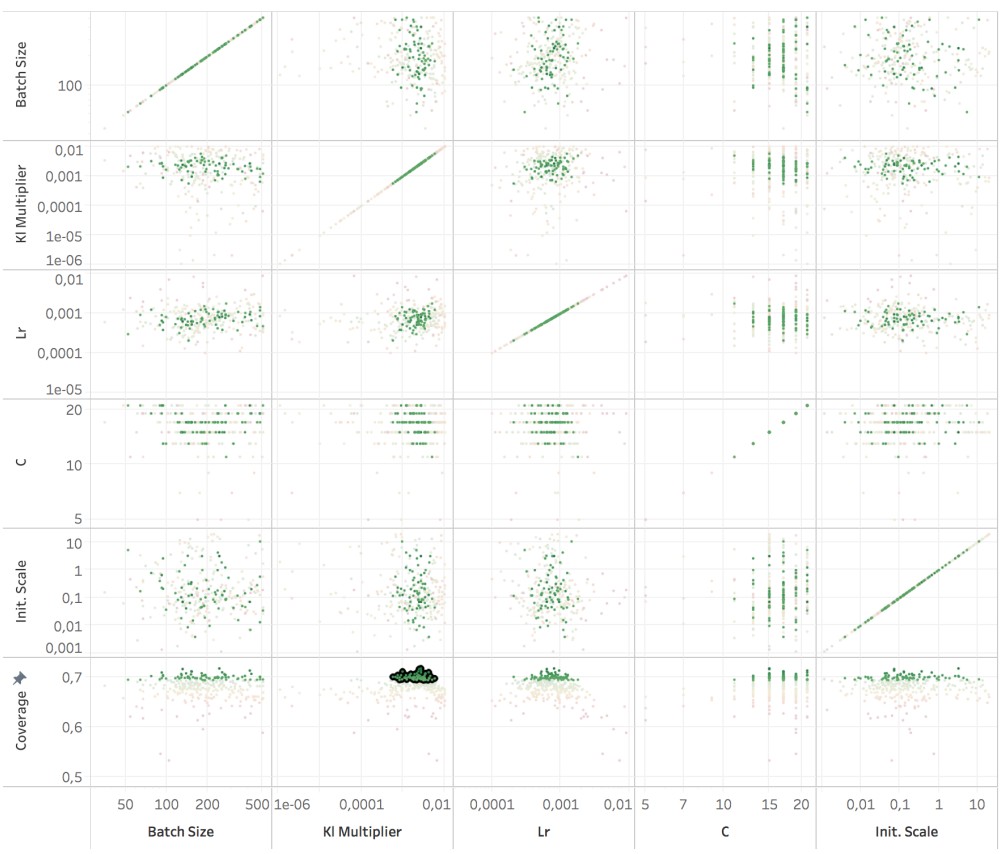

Figure 10: This figure visualizes the pairwise relationship between hyper-parameters of SQUAD and the effect on coverage. The top-60 configurations are highlighted. Green values are good, red values are bad. We have filtered on the optimal settings for bin values and prior to reduce clutter.

