# OpenReview forum: "Predictive Uncertainty through Quantization"
_ICLR.cc/2019/Conference_

### Official Review · AnonReviewer2 · 2018-11-01
**Variational inference with discrete distribution for uncertainty estimation**

**Rating:** 5
**Confidence:** 4

**Review:**

This paper proposes runs variational inference with discrete mean-field distributions. The paper claims the proposed method is able to give a better estimation of uncertainty from the model.

Rating of the paper in different aspects ( out of 10)
Quality 6, clarify 5, originality 8, significance of this work 5

Pros:

1. The paper proposes a generic discrete distribution as the variational distribution to run inference for a wide range of models.

Cons:

1. When the method begins to use mean-field distributions, it begins to lose fidelity in approximating the posterior distributions. Even the model is able to do a good job in approximating marginal distributions, it is hard to evaluate whether the model is gaining benefit overall.

2. I don't see a strong reason for using discrete distributions. In one dimensional space, a distribution can be approximated in different ways. Using discrete distributions only increases the difficulty of reparameterization.

3. In the experiment evaluation, the algorithm seems only marginally outperforms competing methods.


Detailed comments:

In the motivation of the paper, it cites low-precision neural networks. However, low-precision networks are for a different purpose -- small model size and saving energy.

equation 6 is not clear to me.

In equation 10, how are these conditional probabilities parameterized? Is it like: z ~ Bernoulli( sigmoid(wz) ) ?

It is nice to have a brief introduction of the evaluation measure SGR.

In table 3, 1st column, the third value seems to be the largest, but the fourth is bolded.

---

> ### Author Response · Authors · 2018-11-23
> **Response for reviewer 2**
>
> Dear Reviewer 2,
>
> Thank you for your time, comments and suggestions! We think that you'll find your concerns appropriately addressed, and we are looking forward to hear your thoughts.  I'll address them line-by-line below:
>
> "1. The paper proposes a generic discrete distribution as the variational distribution to run inference for a wide range of models."
> The model we propose is a bit different from leveraging generic discrete distributions. It's not comparable with a categorical latent variable model for example, but rather closer to a mixture-of-diracs, which to the best of our knowledge has not been applied in this context yet.
>
> " Even the model is able to do a good job in approximating marginal distributions, it is hard to evaluate whether the model is gaining benefit overall."
> To make the performance of our model more convincing, we have extended the result tables with accuracy and negative log-likelihood metrics.
>
> "2. I don't see a strong reason for using discrete distributions. In one dimensional space, a distribution can be approximated in different ways. Using discrete distributions only increases the difficulty of reparameterization."
> Approximate distributions with high expressiveness, even in the mean-field scenario, are still an open field of research. We explore the avenue of using quantization of the continuous real line to allow a more flexible distribution different form conventional exponential family distributions used in the field, and show not only that it possible to train such a model with competitive performance, but that the expressiveness is used.
>
> "3. In the experiment evaluation, the algorithm seems only marginally outperforms competing methods."
> Our goal is not to reach state of the art results, but to show a novel new model that provides a variety of benefits over existing models. We believe the community benefits from exploring a broad range of models, and not just pursue the single angle that currently seems to lead to the strongest performance. There is still room for improvement in this domain, but we believe that the current results are promising enough to communicate to the community.
>
> "In the motivation of the paper, it cites low-precision neural networks. However, low-precision networks are for a different purpose -- small model size and saving energy."
> I believe that low precision networks have a variety of use-cases, not just small model size. Previous work has shown that low precision networks can outperform their high precision counterparts. It can also serve as an information bottleneck which potentially forces the model to throw away nuisance factors. In our case, we use quantization to make our proposed method tractable, and show that this does not degrade performance
>
> "equation 6 is not clear to me."
> We have improved the clarity of section 3, does this clear things up for you?
>
>
> "In equation 10, how are these conditional probabilities parameterized? Is it like: z ~ Bernoulli( sigmoid(wz) ) ? "
> The individual $p(z_l|.)$ are parametrized as in equation 4 in the updated paper.
>
>
> "It is nice to have a brief introduction of the evaluation measure SGR."
> Agreed, we have extended this introduction and have provide some intuition on its use.
>
> "In table 3, 1st column, the third value seems to be the largest, but the fourth is bolded."
> Thank you for pointing this out, we have updated the results since and reformatted the table.

---

### Official Review · AnonReviewer1 · 2018-11-05
**Too many moving parts**

**Rating:** 4
**Confidence:** 4

**Review:**

The authors consider uncertainty estimation in deep latent variable models. They propose to use quantised latent variable and argue that this solves the overconfidence problem, commonly encountered in variational inference. The proposed approach relies on optimizing an information bottleneck objective instead of  the ELBO.

While the approach is of interest, a number of questions, central to the work, remain. For example, it is not clear how parameter \beta is chosen/optimised, how the number of bins C is chosen and how the annealing scheme is tuned. The authors do not discuss the quantisation parameters, such as bin size and location, which are likely to have a major effect on the performance (and the complexity). Then the authors propose to use a hierachical set of latent variables without properly justifying the need, nor discuss how to select the depth and its impact on the performance. Finally the authors propose yet another extension based on a matrix-factorization with little justification.

Overall, this paper does not fully develop the ideas proposed in the paper or discuss them in sufficient detail. The experiments do not provide additional intuition on what's going on and why this helps and are insufficiently documented/made accessible to be convincing. For example, I am not sure what to conclude from experiments that rely on no (or "light") hyperparameter tuning, when the proposed method has many and not discussion is provided about how to set them or how sensitive results are to their actual value. More importantly, the initial claim that uncertainty is better captured relies on SGR, a metric which is not standard and mentioned in passing without being properly defined. The evaluation further depends on a "selective classifier" which is not detailed, but critical to understanding the experiments.

Finally, the presentation of Section 3 could be significantly improved. For example, I would suggest distinguishing the neural network parameters of the encoder and the decoder as well as the encoder and decoder networks.  I would also refrain using notations like "..." or and always specify what is left and right of an equality. Please spell out all abbreviations at least once in the paper and define all important quantities and concepts.

---

> ### Author Response · Authors · 2018-11-23
> **Respone for reviewer 1**
>
> Dear Reviewer 1,
>
> Thank you for your time and comments, they have helped us further clarify our paper. I believe you might find that your valid concerns have been addressed, and we hope to hear your thoughts on these improvements! I will address your comments line by line below:
>
> "The proposed approach relies on optimizing an information bottleneck objective instead of the ELBO."
> As an aside, deriving the ELBO using Jensen's inequality on a conditional log-likelihood $\log p(y|x) = \int_z p(y|z) p(z|x) dz $ leads to a bound $L= E_{q(z|x)} [ log p(y|z) ] - KL(q(z|x) || p(z|x))$. As $p(z|x)$ is not known, this bound can not be trivially optimized. Hence the focus on the information bottleneck objective which is a more natural motivation for the bound we optimize. From this perspective the $q_\phi(z|x)$ normally found in the ELBO, is now replaced with the true posterior $p_\theta(z|x)$ which is learned directly. This shift in what is assumed to be the variational approximation is somewhat mysterious, and I believe this calls for further study, but perhaps not in this work.
>
> "While the approach is of interest, a number of questions, central to the work, remain. For example, it is not clear how parameter \beta is chosen/optimised, how the number of bins C is chosen and how the annealing scheme is tuned."
> We use an extensive hyperparameter optimization scheme using TPE, having run 10.000's of experiments with different hyperparameter configurations for both our model and baselines. We briefly described this in section 5's penultimate paragraph due to space constraints, but we have extended this description in the paper. We believe this is about as rigorous as comparisons between models can get.
>
> "The authors do not discuss the quantization parameters, such as bin size and location"
> We discuss this in figure 5, and use the hyperparameter optimization scheme to find the optimal configuration.
>
> "Then the authors propose to use a hierarchical set of latent variables without properly justifying the need, nor discuss how to select the depth and its impact on the performance."
> This is a fair point. We were under the impression that using a hierarchical set of latent variables to create a deep latent variable model was a commonly accepted approach to improve models. We have done some initial experimentation where we found that stacking these latent variable layers improve performance, but we believe that to add a rigorous experiment, under the expensive hyperparameter optimization scheme used in the rest of the paper, would be lead to diminishing returns.
>
> "Finally the authors propose yet another extension based on a matrix-factorization with little justification."
> We elaborated on the motivation for this extension in section 5.2. We have moved some of the argumentation further up in the hopes of clarifying this extension.
>
> "Overall, this paper does not fully develop the ideas proposed in the paper or discuss them in sufficient detail. The experiments do not provide additional intuition on what's going on and why this helps and are insufficiently documented/made accessible to be convincing. "
> We have strived to clarify the experiments, and have extended them with more conventional metric results on Negative log-likelihood and accuracy, where we show convincing performance as well.
>
> "For example, I am not sure what to conclude from experiments that rely on no (or "light") hyperparameter tuning"
> Thanks for pointing this out, we have adjusted the wording to clarify. In summary: the notMNIST experiments uses the hyperparameters found on fashionMNIST to evaluate if these hyperparameters are simply overfitted on the dataset specifics, and we find that his is not the case. The SVHN 'light' hyperparameter optimization uses a smaller range of hyperparameter values informed by the findings on the fashionMNIST experiment.
>
> "More importantly, the initial claim that uncertainty is better captured relies on SGR, a metric which is not standard and mentioned in passing without being properly defined"
> To address this we have both included NLL as an extra metric to show the effective uncertainty estimation, and elaborated on the details of SGR and why we use this method.
>
> "Finally, the presentation of Section 3 could be significantly improved."
> Great points, we have incorporated these in the updated paper, please take a look at let me know what you think.

---

### Official Review · AnonReviewer3 · 2018-11-09
**Interesting topic but contributions are not well-motivated**

**Rating:** 5
**Confidence:** 3

**Review:**

The authors propose “Stochastic Quantized Activation Distributions” (SQUAD). It quantizes the continuous values of a network activation under a finite number of discrete (non-ordinal) values, and is distributed according to a Gumbel-Softmax distribution. While the topic is interesting, the work could improve by making more precise the benefit of (relaxed) discrete random variables. This will also allow the authors to more precisely display in the experiments why this particular approach is more natural than other baselines (e.g., if multimodality is the issue, compare to a mixture model; if correlation is a difficulty, compare to any structured distribution such as a flow).

Derivation-wise, the method ends up resembling Gumbel-Softmax VAEs but under an information bottleneck (discriminative model) setup rather than under a generative model. Unfortunately, that in and of itself is not original.

The idea of quantizing a continuous distribution over activations using a multinomial is interesting. However, by ultimately adding Gumbel noise (and requiring a binning procedure), the resulting network ends up looking a lot like continuous values but now constrained under a simplex rather than the real line. Given either the model bias against a true Categorical latent variable, or continuous simplex-valued codes, it seems more natural as a baseline to compare against a mixture of Gaussians. They have a number of hyperparameters that make it difficult to compare without a more rigorous sensitivity analysis (e.g., bin size).

Given that the number of bins they use is only 11, I’m also unclear on what the matrix factorization approach benefits from. Is this experimented with and without?

---

> ### Author Response · Authors · 2018-11-23
> **Thoughts for Reviewer 3**
>
> Dear Reviewer 3,
>
> Thank you for your review! Glad to hear you find our direction interesting and we are grateful for your feedback. I am happy to tell you that most of your concerns are addressed in the (updated) paper, and we have strived to improve the clarity of the writing to make that clear. I'll address your comments line-by-line below, we look forward to hear your thoughts!
>
> "While the topic is interesting, the work could improve by making more precise the benefit of (relaxed) discrete random variables."
> The benefits are tremendous! As stated in the paper, by quantizing the domain of the individual activations and using a categorical distribution over the bins, we can now learn non-linearity's, do not require any batch normalization and fit any mean-field distribution under the quantization scheme, all in a tractable manner.
>
> "compare to any structured distribution such as a flow)"
> A fair point, but as far as I know normalizing flows have not been used in the context of feed-forward prediction, and are rather expensive. We have instead chosen to compare with  baselines that match the computational and application complexity of our model. This would definitely be interesting to study in further work.
>
> " if multimodality is the issue, compare to a mixture model"
> Applying a mixture of Gaussian posterior to the Information Bottleneck objective is non-trivial. It is out of scope to make a fair comparison without previous work paving the way for determining an effective configuration. There are no analytic solutions for the KL between two mixture of Gaussian's, and sampling from the mixture is non-trivial too. We would be interested to see how our model compares against a mixture of Gaussian's posterior, but this more suitable for further work, rather than as a baseline comparison in this paper.
>
> "They have a number of hyperparameters that make it difficult to compare without a more rigorous sensitivity analysis (e.g., bin size)."
> To prevent the effect of these hyperparameters, we have performed an extensive hyperparameter optimization scheme on both our models and the baselines. This is part of the reason why the extra baselines you request are a rather non-trivial extension of this paper. A sensitivity analysis of the hyperparameters was planned but did not make the paperlimit cut. We believe that the experiment presented in table 2 -- using the hyperparameters found on fashionMNIST on notMNIST -- instead present more convincing evidence that the method is robust. Figure 10 in the appendix provides further insight into the pairwise effect of hyperparameters on the coverage.
>
> "the resulting network ends up looking a lot like continuous values but now constrained under a simplex rather than the real line"
> The probability distribution of the values are indeed constrained under a simplex, but the actual latent variable values are constrained by the quantization scheme imposed by the prior. The values v assigned to the 'categories', so to say, are thus ordinal and on the real line. Quantization schemes are inherent to any computational model, we simply increase the quantization noise and incur a small bias penalty of the Gumbel softmax scheme. In return we get a much more expressive distribution under the mean-field constraint.
>
> "Given that the number of bins they use is only 11, I’m also unclear on what the matrix factorization approach benefits from. Is this experimented with and without?"
> In section 5.2 we explain that for the SVHN dataset we explore a larger number of bins, with the optimization scheme finding C=37 with a factorization factor of B=4 as the optimum. We compare against the non-factorized version as shown in table 3. The factorization scheme allows for a higher fidelity of the latent variables, whilst reducing the amount of parameters required per latent variable.

---

### Meta-Review · Area_Chair1 · 2018-12-14
**Not acceptable for ICLR in current form**

**Confidence:** 5
**Recommendation:** Reject

**Metareview:**

The reviewers agree this paper is not good enough for ICLR.